# Action information is integrated into entorhinal representations of conceptual space and is reflected in eye movements

Alexander Eperon[1,2]*, Christian F. Doeller[2,3], Stephanie Theves[2,4,☉]*, Roberto Bottini[1☉]*

**1** Center for Mind/Brain Sciences (CIMeC), University of Trento, Trento, Italy, **2** Department of Psychology, Max Planck Institute for Human Cognitive and Brain Sciences, Leipzig, Germany, **3** Kavli Institute for Systems Neuroscience, The Egil and Pauline Braathen and Fred Kavli Center for Cortical Microcircuits, Jebsen Center for Alzheimer's Disease, Norwegian University of Science and Technology, Trondheim, Norway, **4** Minerva Research Group Neural Codes of Intelligence, Max Planck Institute for Empirical Aesthetics, Frankfurt am Main, Germany

☉ Shared senior author.
* alexander.eperon@unitn.it (AE), stephanie.theves@ae.mpg.de (ST); roberto.bottini@unitn.it (RB)

## Abstract

The hippocampal-entorhinal system represents relations between states in spatial and nonspatial cognitive maps. Critical to understanding how these memory representations are used for cognition is to determine whether the actions underlying state transitions are incorporated in entorhinal cognitive maps. Participants learned to transition between states using different actions, operationalized as mathematical operations. We found that the entorhinal cortex represented the afforded actions across the states. This action representation was not explained by other properties of the task space, such as link distance between the states or reaction times. Furthermore, gaze behavior reflected the direction of afforded actions in the horizontal axis, and the strength of this lateralization predicted both performance and entorhinal pattern similarities, suggesting a link between gaze behavior and neurocognitive mechanisms for navigating conceptual spaces. In sum, this study provides first evidence for the integration of action information into ocular and entorhinal representations of conceptual spaces, suggesting that these may not just map out experiences, but provide information about how to explore knowledge.

## Introduction

Human representations of space are closely connected to behavior. Whereas early maps tended to highlight topography useful to hunter-gatherers, modern mapping apps accentuate the differences between major roads and cycle paths. In light of this, how does the brain support the interplay between spatial representations and action?

The mammalian hippocampal formation is known to contain a variety of spatially-tuned cell types which respond selectively to locations in space [e.g., 1–3] and are

**Data availability statement:** The data and code underlying our results are now publicly available on Zenodo: https://doi.org/10.5281/zenodo.19209884. The code is available on Zenodo, and can be accessed directly from Github at: https://github.com/alexeperon/action-hippo.

**Funding:** S.T.'s research is supported by a Minerva Fast Track Fellowship of the Max Planck Society. C.D.'s research is supported by the Max Planck Society and the Kavli Foundation. A.E.'s PhD position was funded by the Italian Ministry of University and Research (MUR-FARE, MODGET R18WJMSNZF), attributed to R.B. This work was supported by the Italian Ministry of University and Research (MUR–FARE, MODGET R18WJMSNZF) and by the European Research Council (ERC Starting Grant NOAM 804422; ERC Consolidator Grant ATCOM 10112565), attributed to R.B. The funders had no role in study design, data collection and analysis, decision to publish, or preparation of the manuscript. None of the authors received a salary from any of the funders listed above.

**Competing interests:** The authors have declared that no competing interests exist.

**Abbreviations:** AIC, Akaike Information Criterion; BOLD, blood-oxygen-level-dependent; FD, framewise displacement; GE-EPI, gradient-echo-planar imaging; GLMs, general linear models; ILVs, individual-level variables; ISI, Inter-stimulus interval; MPMs, maximum probability maps; NBDA, network based diffusion analysis; neural RDMs, Neural pattern dissimilarity matrices;OADA, order of acquisition diffusion analysis; SRI, simple ratio index; WAIC, widely applicable information criteria.;

thought to offer a neural substrate for a cognitive map [4,5]. Recent evidence has supported the idea that the hippocampal formation encodes relations in support of memory, across spatial and nonspatial domains [6]. For instance, studies have observed that during transitions in two-dimensional feature spaces the entorhinal signal shows a hexa-directional modulation by the trajectory angle [7–14], mirroring the grid pattern observed during spatial navigation [15]. In line with a role in navigation of feature spaces, the hippocampal formation appears to have a role in learning and updating of conceptual knowledge in the hippocampus [16–19]. Moreover, hippocampal representations reflect underlying stimulus structure [20], distances between positions in abstract spaces, such as link distance on a graph [21–24], distances in conceptual and social spaces [10,25–27], or location along a sound frequency spectrum [28].

To exploit such relational knowledge representations, however, we need to know how to move between different states. For instance, to win a chess game, we need to know which actions are afforded at any given state of the game in order to move the pieces from the initial setup to checkmate. In physical navigation, perceptual, vestibular, and motor signals provide the direction and speed information necessary to update the agent's position in the entorhinal network [29]. Although there is no clear nonspatial analogue for this process, domain-general models of the hippocampal-entorhinal system often use reinforcement learning terminology to describe how 'actions' are used to change between 'states' [30]. Accordingly, different models of cognitive maps integrate action input as the cue to move between different states of learned graphs [31–33] or a pre-structured Euclidean space [34]. However, whether and how action information is actually incorporated in relational memory representations remains to be established empirically.

In line with prior work on relational learning and memory, we use the term 'action' to refer to an operation which changes between states [30,35,36], and 'affordance' to refer to the set of actions associated with a state, with no necessary motor component [37–40]. Importantly, the same action can be afforded (and performed) across different points of the space. That is, actions are factorized with respect to state identity and therefore can be learned and used in a state-independent manner [41,42]. Here we focus on the entorhinal cortex, which has been proposed as a neural substrate for the abstraction and generalization of task structure, including action information [39]. Previous work has shown that entorhinal cortex representations generalize task structure across different stimulus sets [43,44], and modeling work has suggested that the entorhinal cortex learns the actions afforded by these structures [32], potentially as part of a larger system for compositional planning [42]. Nonetheless, experimental evidence for this proposal is lacking. Therefore, we investigate if and how the entorhinal cortex represents action information within learned relational structures.

In the context of spatial navigation, although the exact nature of action representation is unclear [45], there is increasing evidence that entorhinal representations are sensitive to changes in idiothetic signals such as speed [46] and gaze and head position [47], as well as behaviorally-relevant environmental changes. Moreover,

entorhinal cells are sensitive to changes in landmarks or goal position [48–51]. This has also been observed in human neuroimaging studies, which have shown distortions of grid-like signatures in fMRI based on goal locations [52], arena layout [53], or prior levels of visual experience [54]. These distortions might be interpreted as a representation of 'action-relevant space' [45,55]. Moreover, incorporating a condition of 'actionability' (that is, representations must enable actions to be carried out) appears to improve on existing models of grid cells by replicating key features such as multiple modules [41]. Nonetheless, it remains an open question if actions themselves are represented in the entorhinal cortex independently of specific stimulus features or environmental layout.

Movement in conceptual spaces has also been shown to be reflected in eye movements, possibly indicating shifts of attention [56, 57,14]. Eye movements also elicit similar representations in visual space to navigable space in the entorhinal cortex [57–62], and hippocampal place cells in chickadees code for the same location whether visited physically or visually [63]. Accordingly, we also used eyetracking to evaluate directional eye movements in our task and their relation to neural representations.

In the current study, we operationalized states as numbers and actions as numerical operations. Participants were taught state-action associations as part of an interconnected graph, and we predicted that states with more similar action associations would elicit more similar entorhinal neural patterns. In sum, we investigated the neural representation of actions using fMRI and eyetracking while participants evaluated numerical operations (actions) afforded by given numbers in a repeating graph structure (states). We found actions to be represented in eye movements and entorhinal cortex activation patterns.

## Results

We predicted that if the entorhinal cortex represents action in a conceptual space, then neural pattern similarities would scale with the action possibilities afforded by different states. We chose to operationalize states as numbers, and actions as numerical operations. This ensured a uniformity of action across different states.

In brief, the experiment consisted of two phases. Firstly, participants were taught a state-action graph in which numbers were associated with numerical operations. Secondly, the representation of individual numbers was probed using eyetracking (on the first day) and then fMRI (on the second day).

### Learning and generalization of state-action associations

Firstly, participants were taught a repeating graph consisting of states and actions projected on a number line. Specifically, they learned that certain numbers permitted transitions to other numbers, according to a predetermined graph structure consisting of repeating modules (see Fig 1c). They learned this through free exploration along the graph using button presses (Fig 1b). Each of the four states afforded a certain subset of actions: state 1 afforded −2 or +2, state 2 afforded +1 or +2, state 3 afforded −2 or −1, and state 4 afforded −2 or +2. In the task, therefore, if the number '49' was mapped onto state 2, participants would be given a choice of changing to numbers '50' or '51' via a left or right button press. Number-state mappings varied across participants by shifting the graph structure along the number line. Therefore, while one participant learned that '49' (state 2) could transform into '50' or '51', another learned that '50' (state 2) could transform into '51' or '52'. These constituted a form of nonspatial affordance: each number was associated with a specific subset of actions based on state assignment.

Knowledge of the graph was tested using blocked test trials presenting two candidate successors, of which only one corresponded to a possible transition and had to be chosen (Fig 1b top right). This tested if participants had correctly learned the afforded actions.

Participants were trained on two consecutive days to learn how specific numbers could 'transform' into other numbers. By the end of the first day, participants were able to reliably select the correct transitions for different numbers (accuracy: 84.7%; chance level 50%, std. 7.1%, last 10 blocks; Fig 2a). Moreover, they were able to apply the learned transformations to new numbers: after training on the second day, they reached a performance level of 88.8% on unseen numbers

 

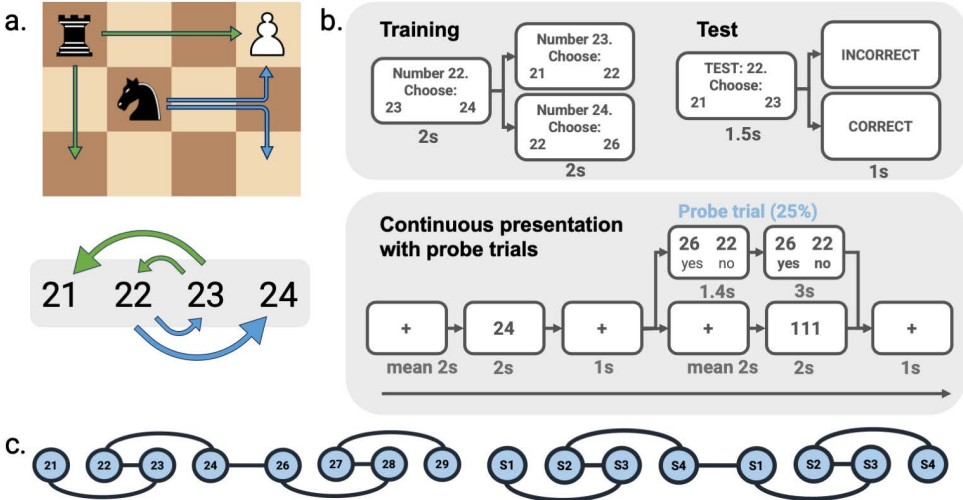

**Fig 1. Experimental design. a.** Representations of (conceptual) space may benefit from the integration of action affordances, e.g., in a game of chess, where information about how pieces can move is necessary to play the game. We tested this idea using numerical operations, whereby participants moved between numbers following the rules of an underlying graph structure. In this example, states 1–4 correspond to numbers 21–24. **b.** Participants learned how numbers may 'transform' into other numbers using an exploration task, whereby they were presented with an individual number and were offered a free choice of two possible successors. When a successor was chosen, the next trial offered the successors for the chosen number. In inter-leaved test blocks, participants indicated which were the 'correct' successors for each number. In the eyetracker and fMRI scanner, individual numbers were presented in a pseudorandom order. To ensure attention on the learned actions, one quarter (16) of trials were followed by probe trials in which participants indicated if two presented numbers were the correct successors. **c.** Unbeknownst to participants, number-action bindings formed part of a repeating sequence arranged in graph 'modules' of four numbers. This can be seen as a set of four states with the same action affordances, projected on a number line. In the example presented, numbers 21 and 26 correspond to state 1 ('*S1*'; and therefore also 31, 36, etc.), but this was assigned differently to each participant. As the modules repeated every five numbers, the structure allowed easy inference of action possibilities for any new numbers. Chess piece icons adapted from a public-domain (CC0) vector source.

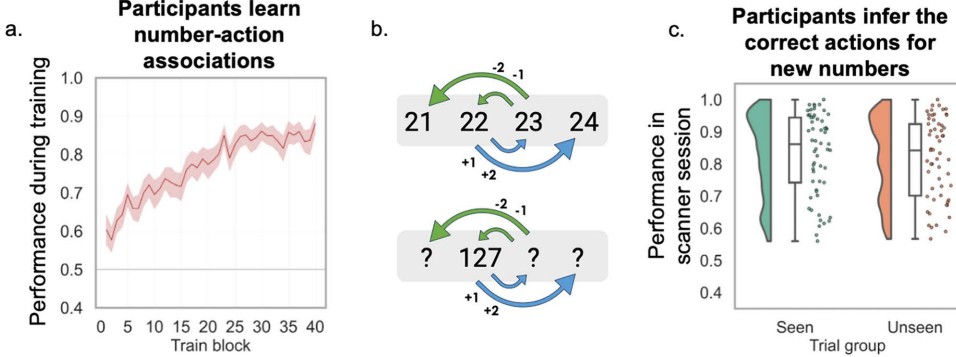

**Fig 2. Behavioral performance. a.** Participants learned the numbers and successors well, achieving a performance of 84.7% by the end of the first day. **b.** The repeating structure should allow participants to infer the correct successors for previously-unseen numbers. **c.** In the recall task in the fMRI scanner, participants were able to correctly identify the successors when presented with unexpected probe trials (mean performance 82.7%). There was no difference in performance between previously-seen numbers and new numbers in the scanner task ($p = 0.126$). All data and code underlying this figure are publicly available on Zenodo: https://doi.org/10.5281/zenodo.19209884.

(std. 11.3%). The application of these operations to hitherto-unseen numbers indicates that participants understood that numbers and transitions were factorized.

In two subsequent sessions using the eyetracker and in the MRI scanner, respectively, participants were then presented with individual numbers in a pseudorandom order. In these sessions their knowledge of the graph was probed using probe trials showing two numbers, to which participants had to respond if both were the correct successors for the previously presented number or not (Fig 1b bottom). This task was designed to extract a representation of neural or ocular signal for each number, which would then be used to test our main hypothesis.

In the continuous presentation task on the first day (during eyetracking), performance was lower than in the exploration task (accuracy: mean 68.1%, std. 15.9% across day 1; missed responses: mean 8.4%, std. 6.4%), but nonetheless improved as familiarity with the task increased to 71.5% in the fourth and final block (std. 21.6%; difference between fourth and first block T = 218.5, p = 0.006, Wilcoxon signed rank test). Participants were better at answering correctly for seen than unseen numbers (seen: mean 70.8%; unseen: mean 65.8%; t(56)=2.78; p = 0.008).

Knowledge of the structure persisted to the second day, suggesting a long-term encoding of the graph information. In the MRI scanner, performance was at a mean accuracy of 82.7% during probe trials (std. 12.5%; missed responses: mean 7.8%, std. 6.3%). Moreover, the differences between seen and unseen numbers diminished to a nonsignificant level, suggesting that longer-term learning was dominated by the abstract structure rather than number-specific action bindings (no significant difference between seen and unseen numbers: seen: mean 83.2%, std. 12.8%; unseen: mean 82.2%, std. 12.3%; t(56) = 1.56, p = 0.13; see Fig 2c).

A stimulus-independent encoding of the graph structure was supported by state-specific reaction time differences during probe trials. Reaction time data in the recall task on the second day showed a difference between response times for different states, comparing participant-wise mean reaction time for each state: state identity predicts reaction times (F(3, 168) = 9.77, p < 0.0001). Post-hoc pairwise comparisons show a significant difference between states 1 and 2 (t(56)=4.44, p < 0.0001), states 1 and 3 (t(56 = 3.45, p = 0.0011), states 2 and 4 (t(56)=−4.09, p = 0.0001) and states 3 and 4 (t(56)=−3.38, p = 0.0014). There were no significant differences between states 1 and 4 (t(56)=0.17, p = 89) and states 2 and 3 (df(56)=−0.23, p = 0.82). Despite these differences in reaction time across states, state identity did not significantly predict performance on probe trials, quantified as the percentage of correct responses following each state (F(3, 168)=2.24, p = 0.085).

Questionnaire data indicated a mix of strategies (specifically, 42.1% reported memory-based strategies, 24.5% relied on calculation and the remaining 33.3% used both), and no participants were aware of the experimental question.

In short, participants were able to learn the numerical operations afforded by different numbers, and apply this knowledge to new numbers. This indicated knowledge of a repeating structure consisting of state-action bindings.

## Eye movement direction reflects the sign of afforded actions

We expected that participants would learn that transitions were similar or different based on mathematical relations between numbers. For this to be the case, they needed to engage with the numerical features of the task. Prior work has suggested that exploration of conceptual spaces is reflected in eye movements, through a lateralization of responses to numerical stimuli [14,56]. Accordingly, we predicted that if participants engaged with the numerical features of our task, eye movements would skew left or right based on the type of actions allowed. Eye movements could be expected to skew more rightwards for states that allow only positive actions (e.g., state 2) relative to states that allow only negative actions (e.g., state 3).

We compared the relative gaze position difference for these states for each time point in a [−0.5, 2.5] window around stimulus onset and tested for each time point in the trial if movement shift was greater than 0 along the horizontal axis, before running cluster correction to account for multiple comparisons. We found one cluster where this was the case (t(45)=5.49, p < 0.0001, cluster window of 899–1,757 ms after onset; Fig 3b). Finally, we show that ocular responses were

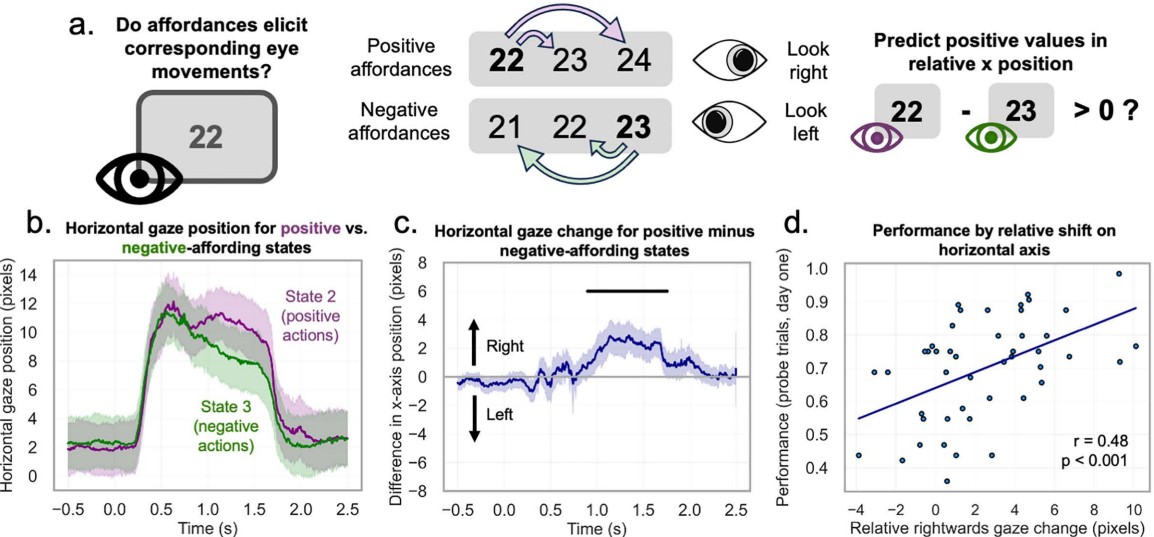

**Fig 3. Ocular representation of afforded actions. a.** We predicted that, if participants were engaging with numerical information, eye movements would skew further right for numbers which afford positive actions, relative to numbers which afford negative actions. **b.** Relative gaze positions for states 2 and 3. When the mean x position of state 3 (which is mapped to number 23 in the example) was subtracted from the mean position for state 2 (mapped to number 22), we find a positive value, indicating a more positive shift for more positive actions. Horizontal bars indicate significance. **c.** The extent of the effect (median position change during the cluster window) correlated with performance in the probe trials of the recall task during eyetracking. Eye icons adapted from a public-domain (CC0) vector source. All data and code underlying this figure are publicly available on Zenodo: https://doi.org/10.5281/zenodo.19209884.

skewed more strongly left-or-rightwards in better-performing participants ($r = 0.478$, $p = 0.00078$, $n = 46$; Fig 3d). This indicates that a stronger (horizontal) schematic implementation may be useful to carrying out the task.

In sum, the behavioral results show that participants had learned and generalized actions across different instances (e.g., new numbers) of the states. Their reflection in eye movements is congruent with engagement with numerical information to solve the task, possibly reflecting use of a mental number line. Thus, next we approached our main question of whether the entorhinal cortex would represent states in terms of the actions they afford.

### Entorhinal pattern similarities scale with afforded action similarity

The main prediction was that the entorhinal cortex would represent affordances in a conceptual space; that is, neural responses would be more similar when states allowed similar actions. To test this, we used representational similarity analysis to compare neural pattern similarity across the states to a model RDM indicating the number of shared afforded actions between states. We refer to this model as an 'affordance model', as it compared states in terms of afforded actions. We found a representation of affordances in the right entorhinal cortex (right: $t(56)=2.13$, $p = 0.019$; left: $t(56)=−0.49$, $p = 0.69$). We further assured that this result is not driven by other effects related to affordance magnitude (the unsigned magnitude of afforded actions) or link distance (the graph distance between states), by including these alternative model RDMs (see Materials and methods; see model RDMs in Fig 4c) as covariates and replicating the finding (right: $t(56)=2.29$, $p = 0.013$; left: $t(56)=−0.60$, $p = 0.73$; Fig 4a–4c). It should be noted that the affordance magnitude model is perfectly anticorrelated with a model based on the number of shared successor states, and therefore this analysis also accounts of any effects of next state prediction. As a further control analysis to check for the specificity of the effect within the medial temporal lobes, we also tested our affordance model in the hippocampus and parahippocampal gyrus, controlling for link distance and affordance magnitude. We found no evidence for an affordance representation in these

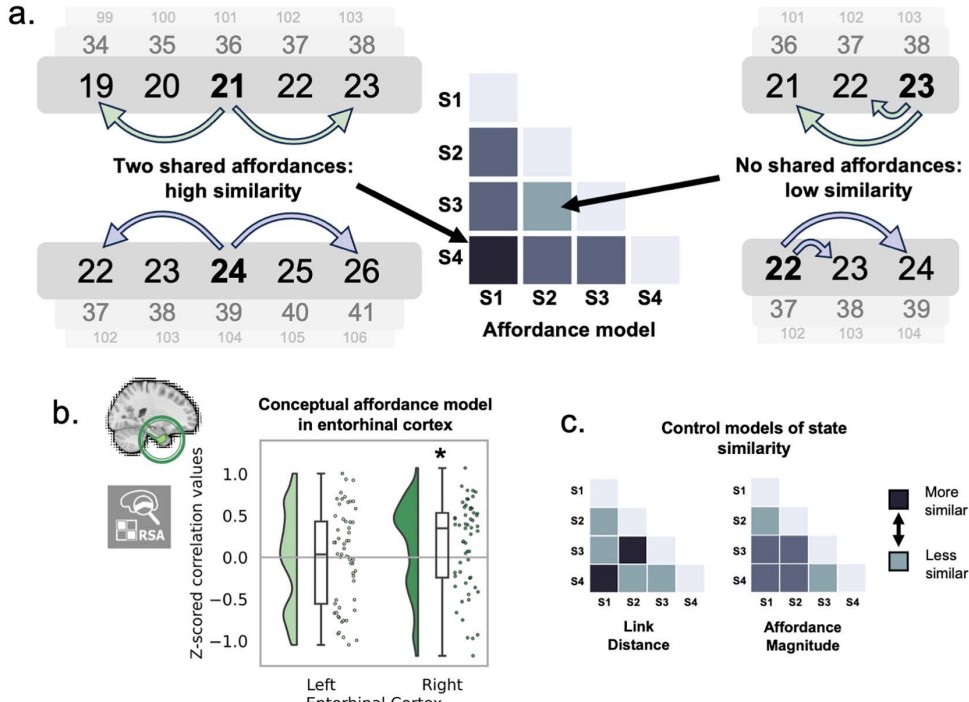

**Fig 4. Neural representation of affordances in a conceptual space. a.** We used representational similarity analysis to compare neural pattern similarity across states in the entorhinal cortex with a model of state similarity based on the number of shared affordances. **b.** Neural pattern dissimilarity matrices (neural RDMs) were calculated in an a priori *region-of-interest* (*ROI*) using cross-validated Mahalanobis distance. Neural patterns in the right entorhinal cortex were correlated with the affordance RDM. **c.** Control models of state similarity. Asterisks indicate significance. All data and code underlying this figure are publicly available on Zenodo: https://doi.org/10.5281/zenodo.19209884.

regions (hippocampus left: t(56)=1.24, *p* = 0.11; hippocampus right: t(56)=1.59, *p* = 0.059; parahippocampal gyrus left: t(56)=−1.01, *p* = 0.84; parahippocampal gyrus right: t(56)=0.27, *p* = 040).

To further confirm the stability of the entorhinal affordance representation, we confirmed that cross-validated mean distances in the entorhinal RDMs were greater than 0, indicating our neural RDMs reflected consistent differences in neural responses to different conditions across runs. The affordance effect persisted within a subset of subjects whose mean RDM values were greater than 0, indicating that for subjects in which entorhinal voxels respond consistently to different states, neural patterns correlate with the predicted affordance model. (Fig A in S1 Appendix). Moreover, we used reliability-based voxel selection [64] to sub-sample voxels within our ROIs which responded consistently across different runs. Likewise, we replicated the affordance representation in the entorhinal cortex, indicating that it is driven by reliable, i.e., more condition-responsive, voxels (Fig B in S1 Appendix). Additionally, the effect is within the expected noise ceiling in the right entorhinal cortex, and therefore as strong as it could be given noise (Fig C in S1 Appendix). Finally, we demonstrate that the effect is also not driven by potential reaction time differences between the states using a partial correlation analysis (Fig D in S1 Appendix).

Next, we ran a whole-brain searchlight to establish the spatial distribution of the affordance effect. For each participant, each voxel was taken as the center of a sphere, which was used to run representational similarity analysis as before. Whole-brain maps were then transformed to MNI space, and one-sided t-tests were used to compare group-level correlation scores at each voxel to zero. However, no voxels survived cluster correction (threshold-free cluster enhancement; all *p* > 0.1).

Finally, we also tested for an affordance representation in eye movements during the fMRI session on day 2, corresponding to our finding on day 1. Since we did not perform eyetracking in fMRI, we used DeepMReye, a toolbox based on a convolutional neural network (CNN) [65] to predict gaze position from functional MRI data at a sub-TR resolution. We tested whether participants would look further right for positive- as compared to negative-affording states in the previously-identified cluster window (see Materials and methods) and again find that participants' estimated gaze position tended to be further right for positive-affordance numbers than for negative-affording numbers (paired $t$ test; t(56)=2.0279, $p$=0.024; Fig 5A and 5B). Using this fMRI based estimate, the correlation to performance no longer reached significance ($r$=0.194, $p$=0.1482).

We would expect that participants with more pronounced gaze differences would also show greater neural distances in the entorhinal cortex between conditions eliciting lateralized eye movements Fig 5A and 5B. Regarding the relation to the entorhinal representation, we found that the (normalized) neural distances between states 2 and 3 correlated with the horizontal distance between median gaze positions ($r$=0.263, $p$=0.0484, $n$=57, Fig 5C). Accordingly, we also observed a trend in the correlation between the strength of the entorhinal affordance representation to the magnitude of the gaze lateralization (states 1 and 4; $r$=0.231, $p$=0.0844; $n$=57). These may indicate a relation between entorhinal representations of conceptual information and gaze behavior.

To ensure that the entorhinal cortex representations were not driven by simple retinotopic changes induced by eye movements, we created an alternative model based on gaze position in 2D space. To wit, we took the euclidean distance between the median x and y position for each condition (across trials; for details see Materials and methods, Supplementary Materials). We validated this model by showing that it was represented in a visual cortex ROI (t(56)=2.119; $p$=0.0193). Nonetheless, this model was not represented by the right entorhinal cortex (t(56)=−0.771, $p$=0.778), and the affordance effect persisted when this visual model was excluded in a partial correlation (t(56)=1.882, $p$=0.0369, see Fig E in S1 Appendix). This suggests that while our affordance model is connected to gaze behavior, it cannot be explained with reference to eye movement alone.

In sum, we find that actions (i.e., possible numerical operations) are represented in the entorhinal cortex and indirectly reflected in gaze behavior. This entorhinal affordance representation was not driven by alternative properties such as link distance or magnitude, or by general gaze behavior, and is robust with respect to selected participants, voxels, and noise ceiling.

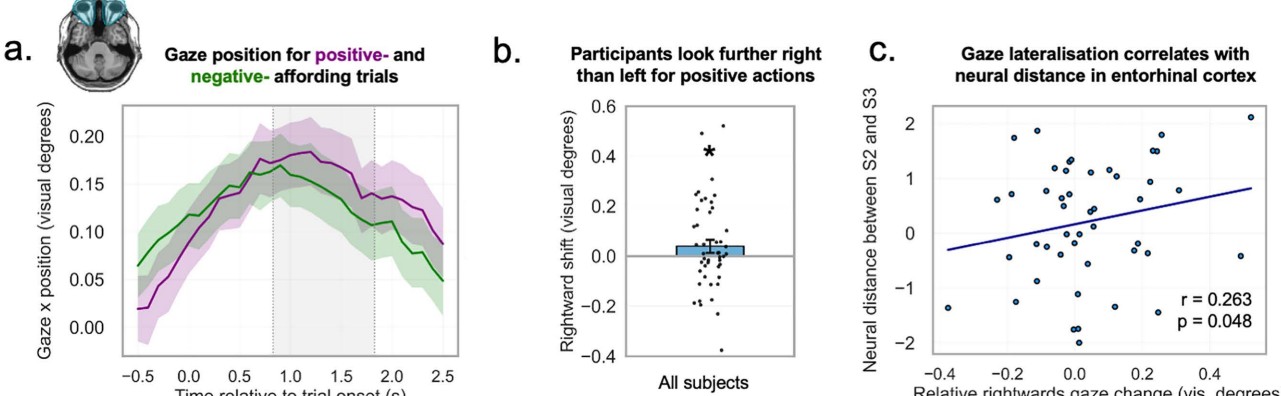

**Fig 5. DeepMReye-based gaze lateralization. a.** Predicted gaze positions during the functional MRI scan were extracted using DeepMReye. We expected that the median gaze position would be further right for positive-affording numbers than for negative-affording numbers during the cluster window identified using eyetracker data from the previous day. Brain image from Frey and colleagues [65]. **b.** The median x position within the cluster window was further right for positive-affording trials (state 2) than negative-affording trials (state 3). **c.** The strength of the gaze lateralization effect correlated with the predicted neural distances in the right entorhinal cortex (after normalization across the RDM) for states 2 and 3. All data and code underlying this figure are publicly available on Zenodo: https://doi.org/10.5281/zenodo.19209884.

                                      

## Discussion

The hippocampal formation has a demonstrated role in mapping out relational knowledge [5,39,66,67] with potential implications for general cognitive performance [68,69]. In the absence of idiothetic signals, however, it is unclear how the brain 'navigates' these memory representations—to play a game of chess, we need to know both the layout of the pieces and how they move. Prior work has demonstrated that the entorhinal cortex represents action-relevant information during spatial navigation, and models of the hippocampal formation often assume a fixed 'action' input to move between states [31,32,70], an abstraction of the velocity input used in models of spatial navigation [29,34,71–77] with no clear neuroanatomical substrate. Here, we provide evidence that the human entorhinal cortex does represent actions in a conceptual space, and that these representations are linked to gaze behavior.

We used a number line as base structure to which states with different afforded numerical operations (actions) were associated, thereby allowing us to distinguish action representations from other features of a state space (e.g., state identity, link distance). We find that the entorhinal cortex represents the actions 'afforded' by a given state in our design. It should be noted, however, that we only test a single conceptual domain: further work will be needed to establish to what extent action representation is domain-general, as suggested by models of the hippocampal formation [70].

In line with previous work linking eye movements to a simulated mental number line [14,52,78,79], afforded actions were also tracked by eye movements: participants looked further rightwards when afforded actions were positive relative to states with negative action affordances. This may show a projection of affordances on a spatial frame, distinguishing our work from prior arithmetic-related findings in the MTL [e.g., 80]. We also show that the gaze lateralization effect during eyetracking is positively correlated with task performance, possibly indicating allocation of attention [57]. A potential functional role of eye movements might also be indicated by their relation to entorhinal affordance representations. Our results would go in line with findings that visual grid cells are modulated by recognition memory [nonhuman primates: [58]; humans: [62] and may extend the proposed link between attention and spatial navigation [81] to conceptual spaces [57].

More generally, our work fits in with a growing body of evidence that entorhinal cognitive maps are tied to learned behavior. Previous fMRI studies have shown that neural representations in the entorhinal cortex and hippocampus are influenced by task demands or the relevance of dimensions and goals [26,82,83]. For instance, when learning new concepts, the hippocampus organizes stimuli according to the category-defining dimensions and not according to all perceptual or mnemonically-relevant dimensions [26]. Blind humans exhibit a four-directional symmetry instead of the typical hexa-directional symmetry likely elicited by grid cell populations, suggesting that entorhinal representations may reflect long-term behavioral patterns [54]. Similarly, rodent and human work has shown a distortion in grid signal as a function of goal placement [48,49,52]. Moreover, rodent intracranial work has shown that grid cells display realignment toward landmarks, which appears to reflect future behavior. These distortions are too slow to explain one-shot learning (the authors invoke a mediating plasticity mechanism as a potential solution) but nonetheless suggest a coupling of entorhinal representations with behavior [51]; for landmark alignment in a mental simulation task see [84; 85]. Our findings generally support the intuition that entorhinal representations are influenced by afforded actions [for a computational model of grid cells based on action constraints, see [41].

This is an intuition shared by models of cognitive maps [39,70], which have suggested that the entorhinal cortex either learns a state-action graph [32] or is biased by the transition statistics of an environment, which are in turn subject to available actions [successor representation; 31]. Our observed action signal may be considered more in line with the first account, as successor representation models do not predict an explicit learning of action information in the entorhinal cortex. Nonetheless, our results may be compatible with distortions in transition structure as a function of different learned actions—indeed, previous fMRI work has shown representations of probabilistic task structure in the entorhinal cortex [43]. As successor representation models are built around a predictive framework, future work may aim to test this by looking at how neural representations of action are related to trial-level choice behavior (which we do not capture) or explicitly controlling reward distributions.

 

There are multiple possible neural substrates for processing of actions in the entorhinal-hippocampal system. In spatial navigation, action is integrated into the grid cell network as velocity input, which may be derived from idiothetic signals or internally generated [29,75,86] and can be described by specialized cell types in the entorhinal cortex [46,47,87]. Intriguingly, Mallory and colleagues [47] identify entorhinal cells which conjunctively code for rodent eye and body position, reminiscent of saccade direction cells in head-fixed primates [59,88]. Beyond spatial navigation, it has been suggested [32,70] that action computations can be carried out through vector cells such as object-vector cells [3] and goal-vector cells [89]. Based on our fMRI data, it is difficult to draw a direct link between action representation and specific neural populations. In light of the role for vision in primates and work suggesting a role of eye movements in 'simulating' conceptual navigation [14,52,57,61,62,81,88], it is conceivable that the observed eye movements reflect an embodiment of action. This could function as action input through cells with responses similar to saccade direction cells [59]. At a more abstract level, eye movements could reflect movements of attention in the conceptual space [14,57,90]. If that is the case, shifts of internal spatial attention (rather than simple eye movements) could provide a velocity signal to the entorhinal system. This might explain why we find a tentative link between (estimated) eye movement and neural representations, but control analyses suggest that our entorhinal signal is not driven purely by eye movement (Fig E in S1 Appendix).

The present results provide evidence that the entorhinal cortex represents actions in a conceptual domain. These findings should lend weight to a closer link between cognitive maps and behavior, in particular gaze behavior. In his seminal work on visual perception, James Gibson eschewed the idea of cognitive maps because of the lack of an "internal perceiver to read [the] map" [91]. Our work suggests that, on the contrary, cognitive maps could provide a guide to navigation through action representation.

## Materials and methods

### Participants

Sixty healthy German-speaking volunteers participated in the study. All participants were right-handed, with normal or normal-to-corrected vision and reported no history of neurological or psychiatric disorders. All participants provided written informed consent to participate in the study, and were compensated for participation. The study was approved by the local ethics committee at the University of Leipzig, Germany (protocol number 437/22-ek), and conducted according to the principles expressed in the Declaration of Helsinki. Three participants did not reach a predetermined performance threshold of 74/128 correct answers in the scanner task, which corresponded to a performance greater than chance in a binomial test at an alpha level of 0.05. These participants were excluded from all analyses. Within the remaining 57 subjects, 29 self-identified as female with an overall mean age of 26.7 (s.d. 4.9). For a further 11 subjects, eyetracking data was incomplete (minimum one full run missing; nine subjects) or inaccurate (all data points further than half of screen width or height from the center during trials; two subjects) due to issues with binocular calibration resulting from thick glasses, contact lenses or makeup. This resulted in a final sample of 46 subjects for ocular analyses (21 self-identified female; mean age 25.8, s.d. 4.6).

The sample size was determined without a prior power analysis, due to the difficulties in making an accurate estimate deriving from the small number of previous related studies and the multiple planned analyses. Initially, a high drop-out rate was expected due to a taxing task involving numbers, and therefore it was decided to recruit 60 participants in order to match or exceed the larger sample sizes typical of studies which have been able to detect comparable effects in the entorhinal cortex [11,23,52].

### Experimental design

The experiment took place over two consecutive days. On the first day, participants were informed that they would play the role of an archaeologist who had discovered a new number system in local sandstone caves whereby numbers could be transformed into other numbers. This was, of course, fictitious and designed to ensure they understood that

 

they should try to learn number transitions from scratch rather than apply any preexisting intuitions. Participants were instructed that their task was to first understand how numbers could 'transform' into other numbers, and then use this knowledge to identify correct number transformations. These transformations would define the 'actions' used in future analyses: if the number 25 could transform into the number 27, for example, we can intuit that 25 affords the action +2.

Each day consisted of both training and a recall task. The training was designed to teach participants how numbers were connected to other numbers by interleaved exploration and test blocks (see below). The recall task was used to measure neural and ocular responses to each number, which would then be tested based on the similarity between affordances.

On the first day, participants carried out the training for approximately an hour and a half (40 blocks), followed by half an hour of a recall task (4 runs) while eye movements were monitored using an eye tracker (details below). On the second day, they repeated the training for only 20 min (10 blocks), followed by an hour of the recall task (8 runs) while brain activity was monitored using fMRI. At the end of the experiment, participants were asked to fill out a questionnaire, which primarily aimed to test strategies used while carrying out the task, and gather relevant demographic information.

## Method details

### Stimuli

Our design aimed to distinguish between previously-tested features of a state space (e.g., link distance) and the specific transitions (numerical operations, referred to interchangeably as 'actions') that could be carried out within this structure.

To dissociate state identity from specific stimulus identity, we created a repeating structure (module) based on relations between four states. This module repeated every five numbers, allowing easy generalization while avoiding visual confounds in the design. Each state (number) was associated with two possible actions (arithmetic calculations: +2, −2, +1, −1) and participants could navigate the numberline by choosing one of them. For instance, if number 23 afforded actions −1/−2, it would be possible to select 21 but not 25 as the next state.

Using a number line as the base structure allowed us to manipulate the transitions independently from the specific states. For example, transitioning between any two numbers could involve a big numerical change but still constitute one link in the graph structure. As previous studies reported a linear encoding of numerical knowledge, we assumed that the relationship between actions would hold independently of the numbers seen (e.g., '3' to '5' would be similar to '55' to '57'). Likewise, while a movement from '3 to '2' may be a single step, it constitutes a different transition (−1) from that of the next step '2' to '4' (transition of +2).

To further control for any pre-learned mathematical associations, we balanced number-state assignments across participants by shifting the position of graph modules along the number line. Whereas one participant may learn the trajectory 25 - 27 - 26 - 28, another might learn 36 - 38 - 37 - 39.

In total, participants were trained and tested on 4 and 8 modules (16 and 32 numbers) respectively prior to entering the fMRI scanner, while each run consisted of an extra 3 previously unseen modules (12 numbers). In total, the available structure seen throughout the entire experiment consisted of 36 modules per participant, corresponding to 144 numbers. All numbers were greater than 20, to avoid interactions with preexisting associations related to single-digit numbers.

### Training

Training was designed to teach participants the transformations between numbers using a 'free exploration' paradigm. Participants were presented with one number on the screen, alongside two options which corresponded to the two directly connected numbers in the graph structure introduced above. Selecting a number from the two choices led to the next display including the selected number with its two direct connections (see Fig 1). This phase can be seen as an exploration of the number line using the numerical operations available in the structure.

After 16 exploration trials, participants were presented with 18 test trials on the same numbers, in which only one of the two options corresponded to a correct successor according to the graph structure. In these trials, participants were presented with a similar screen, consisting of a central number above two possible successors. In this case, however, only one successor was correct. Participants were asked to select the correct successor and were given feedback on the screen after each trial.

After 20 and 40 of these train and test blocks, as well as at the end of the training on the second day, 24 test trials using novel numbers (drawn from the beginning and end of the participant-specific numberline) were presented without feedback. This allowed a test of how well participants applied the learned transitions to new numbers, which we refer to as 'weak generalization' due to visual clues resulting from the length-five graph modules (i.e., if number 29 afforded 27 and 31, number 79 would afford 77 and 81).

Test trials were evenly selected from different positions within stimulus modules. Most trials were selected from within a range of two from the presented number in order to avoid use of a distance-based heuristic. Nonetheless, to ensure participants paid attention to the entire number rather than the last digit alone, each test block included two trials where the potential successors differed from a correct answer by a multiple of 10 (e.g., if 23 could be followed by 25, the number 35 might be displayed).

### Recall Task (eye tracker and MRI)

During eyetracking and the fMRI session participants were presented with a series of numbers, including previously-seen and new numbers. To keep participants engaged in the task, at random intervals probe trials queried their understanding of the affordance-state bindings.

If these two numbers were the two correct successors of the most recent number, participants had to press a button to 'accept' them. Otherwise, they had to reject the option. Half of probe trials showed correct possible successors.

Buttons were counterbalanced across runs, and the ordering of presented probe trial stimuli was randomized, to ensure there was no systematic bias in either motor responses or visual attention. To avoid potential problems of temporal auto-correlation in the MRI scanner, the order of conditions was counterbalanced such that each trial was preceded an equal number of times by trials from different conditions.

Inter-stimulus (ISI) intervals were drawn from a truncated exponential distribution with a mean of 3 s in the MRI scanner and 2.5 s during eyetracking, respectively. To guarantee that the ISI selection did not influence the final results due to temporal autocorrelation, for each run we ensured that the Spearman correlation between ISI length between two trials and model based trial distance was less than 0.1 via permutation of possible ISI sequences across all models (see 'Analysis: Representational Similarity Analysis' for model details).

In the eyetracker, participants carried out four runs of ~6 min each. Each run consisted of 80 trials, of which 16 were probe trials.

In the MRI scanner, participants carried out eight runs, each of which lasted 7 min. As with the eyetracker, each run consisted of 80 trials of which 16 were probe trials. This resulted in a total of 512 trials for analysis.

### MRI Data Acquisition

MRI data were recorded using a 3 Tesla Siemens Magnetom Prisma Fit scanner (Siemens, Erlangen, Germany) with a 32-channel head coil.

Following a localizer scan, blood-oxygen-level-dependent (BOLD) contrast was measured for the eight runs of the recall task using T2*-weighted whole-brain gradient-echo-planar imaging (GE-EPI) with a multi-band acceleration factor of 5.

The fMRI sequence has the following parameters: TR = 1000 ms; TE = 22 ms; voxel size = 2.5 mm isotropic; field of view = 204 mm; flip angle = 62°; partial fourier = 0.75; bandwidth = 1,794 Hz/Px; multi-band acceleration factor = 5; 65 slices interleaved; distance factor = 10%; phase-encoding direction: A -> P. Per run, 415 volumes were acquired.

After every run, fieldmaps were recorded as a way to measure and correct for inhomogeneities in the magnetic field. Fieldmaps were acquired using opposite phase-encoded EPIs, using the same voxel sizes and field of view with TR = 8000 ms; TE = 50 ms; flip angle = 90°; partial fourier = 0.75; bandwidth = 17.934 Hz/Px.

Following recommendations from previous work [92], the acquisition box was manually aligned by the experimenter to line up with the long axis of the hippocampus. This has been suggested as a method to acquire clearer signal in the medial temporal lobes.

After each scanning session, an anatomical scan was recorded. This was done using a T1-weighted MP2RAGE (TR = 5000 ms, TE = 19.6 ms, voxel size = 1 mm isotropic). The resulting scan was denoised using the method outlined in [93].

All stimuli were projected onto a screen situated above the participant using a mirror attached to the head coil. Behavioral responses were collected using an MRI-compatible button box.

### Eyetracker data acquisition

Participants were seated in front of a monitor leaning on a chin rest and forehead bar, at a distance of 57 cm from the screen. We recorded binocular gaze position continuously using an EyeLink 1,000 Plus (SR Research), at a sampling rate of 1,000 Hz. To ensure accurate recording, we calibrated the system at the start of each run using the built-in calibration and validation protocols from the EyeLink software using nine fixation points. Validation was accepted when gaze points were less than one degree from points. Accuracy (mean distance from the screen center) and precision (root mean square distance from mean gaze position) were calculated using a 1 s window before stimulus onset while viewing a fixation cross. Precision had a mean of 12.83 pixels and a standard deviation of 7.97 pixels across subjects, while mean accuracy was 17.20 pixels, with a standard deviation of 6.65 pixels.

### Preprocessing of fMRI data

Preprocessing was carried out using fMRIprep 22.0.1 (see below for a boilerplate provided by fmriprep). The processing steps include segmentation, head-motion estimation and correction, slice timing correction, co-registration to anatomical images, fieldmap-based (AP-PA) susceptibility distortion correction and resampling to MNI space. All analyses were carried out in volumetric space, using subject space (T1w) for all analyses carried out at the single-subject level. Before running general linear models (GLMs) for any analysis, volumetric data were smoothed using a Gaussian FWHM filter with a 5 mm width.

Results included in this manuscript come from preprocessing performed using fMRIPrep 22.0.1 [94, 95]; RRID: SCR_016216), which is based on Nipype 1.8.4 [96,97]; RRID: SCR_002502).

### Preprocessing of B0 inhomogeneity mappings

A total of 8 fieldmaps were found available within the input BIDS structure for each subject. A B0-nonuniformity map (or fieldmap) was estimated based on two (or more) echo-planar imaging (EPI) references with topup [98]; FSL 6.0.5.1:57b01774).

### Anatomical data preprocessing

A total of 1 T1-weighted (T1w) images were found within the input BIDS dataset. The T1-weighted (T1w) image was corrected for intensity non-uniformity (INU) with N4BiasFieldCorrection [99], distributed with ANTs 2.3.3 [100], RRID: SCR_004757], and used as T1w-reference throughout the workflow. The T1w-reference was then skull-stripped with a Nipype implementation of the antsBrainExtraction.sh workflow (from ANTs), using OASIS30ANTs as target template. Brain tissue segmentation of cerebrospinal fluid (CSF), white-matter (WM), and gray-matter (GM) was performed on the

                                                                                 

brain-extracted T1w using fast [FSL 6.0.5.1:57b01774, RRID: SCR_002823, 101]. Volume-based spatial normalization to two standard spaces (MNI152NLin6Asym, MNI152NLin2009cAsym) was performed through nonlinear registration with antsRegistration (ANTs 2.3.3), using brain-extracted versions of both T1w reference and the T1w template. The following templates were selected for spatial normalization: FSL's MNI ICBM 152 nonlinear 6th Generation Asymmetric Average Brain Stereotaxic Registration Model [102], RRID: SCR_002823; TemplateFlow ID: MNI152NLin6Asym, ICBM 152 Nonlinear Asymmetrical template version 2009c [103], RRID: SCR_008796; TemplateFlow ID: MNI152NLin2009cAsym.

## Functional data preprocessing

For each of the 8 BOLD runs found per subject (across all tasks and sessions), the following preprocessing was performed. First, a reference volume and its skull-stripped version were generated using a custom methodology of fMRIPrep. Head-motion parameters with respect to the BOLD reference (transformation matrices, and six corresponding rotation and translation parameters) are estimated before any spatiotemporal filtering using mcflirt [FSL 6.0.5.1:57b01774, 104]. The estimated fieldmap was then aligned with rigid-registration to the target EPI reference run. The field coefficients were mapped on to the reference EPI using the transform. BOLD runs were slice-time corrected to 0.451 s (0.5 of slice acquisition range 0 – 0.902 s) using 3dTshift from AFNI [105, RRID: SCR_005927]. The BOLD reference was then co-registered to the T1w reference using mri_coreg (FreeSurfer) followed by flirt [FSL 6.0.5.1:57b01774, 106] with the boundary-based registration [107] cost-function. Co-registration was configured with six degrees of freedom. Several confounding time-series were calculated based on the preprocessed BOLD: framewise displacement (FD), DVARS and three region-wise global signals. FD was computed using two formulations following Power (absolute sum of relative motions, Power and colleagues [108]) and Jenkinson 2022 (relative root mean square displacement between affines [104]). FD and DVARS are calculated for each functional run, both using their implementations in Nipype [following the definitions by Power and colleagues [108]]. The three global signals are extracted within the CSF, the WM, and the whole-brain masks. Additionally, a set of physiological regressors was extracted to allow for component-based noise correction [CompCor, 109]. Principal components are estimated after high-pass filtering the preprocessed BOLD time-series (using a discrete cosine filter with 128 s cutoff) for the two CompCor variants: temporal (tCompCor) and anatomical (aCompCor). tCompCor components are then calculated from the top 2% variable voxels within the brain mask. For aCompCor, three probabilistic masks (CSF, WM, and combined CSF + WM) are generated in anatomical space. The implementation differs from that of Behzadi and colleagues in that instead of eroding the masks by 2 pixels on BOLD space, a mask of pixels that likely contain a volume fraction of GM is subtracted from the aCompCor masks. This mask is obtained by thresholding the corresponding partial volume map at 0.05, and it ensures components are not extracted from voxels containing a minimal fraction of GM. Finally, these masks are resampled into BOLD space and binarized by thresholding at 0.99 (as in the original implementation). Components are also calculated separately within the WM and CSF masks. For each CompCor decomposition, the k components with the largest singular values are retained, such that the retained components' time-series are sufficient to explain 50 percent of variance across the nuisance mask (CSF, WM, combined, or temporal). The remaining components are dropped from consideration. The head-motion estimates calculated in the correction step were also placed within the corresponding confounds file. The confound time-series derived from head-motion estimates and global signals were expanded with the inclusion of temporal derivatives and quadratic terms for each [110]. Frames that exceeded a threshold of 0.5 mm FD or 1.5 standardized DVARS were annotated as motion outliers. Additional nuisance time-series are calculated by means of principal components analysis of the signal found within a thin band (crown) of voxels around the edge of the brain, as proposed by [111]. The BOLD time-series were resampled into standard space, generating a preprocessed BOLD run in MNI152NLin6Asym space. First, a reference volume and its skull-stripped version were generated using a custom methodology of fMRIPrep. Automatic removal of motion artifacts using independent component analysis [ICA-AROMA, 112] was performed on the preprocessed BOLD on MNI space time-series after removal of nonsteady state volumes and spatial smoothing with an isotropic, Gaussian kernel of 6 mm FWHM (full-width half-maximum). Corresponding "non-aggressively"

 

denoised runs were produced after such smoothing. Additionally, the "aggressive" noise-regressors were collected and placed in the corresponding confounds file. All resamplings can be performed with a single interpolation step by composing all the pertinent transformations (i.e., head-motion transform matrices, susceptibility distortion correction when available, and co-registrations to anatomical and output spaces). Gridded (volumetric) resamplings were performed using antsApplyTransforms (ANTs), configured with Lanczos interpolation to minimize the smoothing effects of other kernels [113]. Nongridded (surface) resamplings were performed using mri_vol2surf (FreeSurfer).

Many internal operations of fMRIPrep use Nilearn 0.9.1 [114, RRID: SCR_001362], mostly within the functional processing workflow. For more details of the pipeline, see the section corresponding to workflows in fMRIPrep's documentation.

Copyright waiver

## Statistical analyses

**Analysis of behavioural data.** Performance was measured during the training tasks as the percentage of correct responses in the test phases (see above). On the first day, this corresponded to a total of 720 trials (change level of 50%). Performance on unseen numbers was measured using the two extended test phases after 20 and 40 blocks, which consisted of 48 trials. On the second day, participants were presented with 180 test trials on learned numbers and 24 test trials consisting of previously-unseen numbers.

In the recall task, performance was measured relative to probe trials, of which there were 16 per run (therefore 64 in the eyetracker and 128 in the fMRI scanner; chance level 50%). Participants had slightly longer to reply in the fMRI scanner (1.4 s instead of 1.2 s).

Half of the probe trials consisted of previously-unseen numbers. To compare performance on seen and unseen numbers, a two-sided paired $t$ test was used with an α level of 0.05.

Differences in reaction time in the recall task were measured using repeated measures ANOVA (α of 0.05) to ask if state identity predicted the mean reaction time for each state, for numbers followed by a probe trial. This was particularly relevant for the second day, to ensure differences in reaction times (and thus potentially task difficulty) could not explain our model of interest. Post-hoc pairwise comparisons were then carried out using paired $t$ tests across combinations of states, using a two-tailed test with a Bonferroni-corrected α of 0.0083 to reflect six pairwise comparisons.

Questionnaire data was read by the experimenter to ensure no participants were aware of the experimental question. To obtain a measure of strategy, the percentage (of 57) who replied 'memory', 'calculation', or 'both' in the multiple choice question "How did you resolve the task today?" was measured.

**Representational similarity analysis.** Representational similarity analysis [115] is a correlational method which compares the (dis)similarity of neural signals in different conditions or trials to a matrix of predicted model based distances.

To run RSA, we first ran a first-level GLM on the preprocessed data in order to extract beta values for each condition. The GLM included motion regressors from fmriprep (see above), as well as an intercept. We also included regressors modeling visual information displayed on the screen and motor actions, specifically separate regressors for each type of visual presentation (stimulus presentation, probe trial presentation, response presentation, and fixation cross) and for left/ right button presses. As conditions, we included the state identity for each presented number, modeled as a stick function.

This model was run separately on each of the 8 functional runs in native (subject) space (T1w), to obtain a map of beta values for each condition, run, and every subject. This allowed us to run cross-validated dissimilarity measures, which

identify signal common to the same condition across runs and thereby reduce the effect of run-wise noise. The cross-validated Mahalanobis distance ['crossnobis'; 116,117] was used, using the residuals from the first-level GLM to estimate the relevant noise covariance matrix for each run.

As the main hypothesis concerned a specific region (namely, the entorhinal cortex), a ROI approach was used. ROIs were sourced from the Jülich Atlas [118, version 3.0.3] in MNI space, taking the maximum probability maps (MPMs) for each region (right entorhinal cortex, left entorhinal cortex, visual cortex (V1 + V2), and finally left and right motor cortex (4a and 4p), parahippocampal gyrus and hippocampus for control analyses). ROIs were then mapped into subject space using ANTs and the transformation matrices output by fmriprep. The resulting ROIs of interest had the following sizes: left entorhinal cortex (mean 111.5 voxels, standard deviation 12.3), right entorhinal cortex (mean 140.1 voxels, standard deviation 22.0).

Based on this, we then calculated a neural dissimilarity matrix (see above) using the four different conditions in the experiment. This was then compared to model matrices (see below) using a tie-corrected spearman rank correlation, resulting in a single correlation value. Partial correlation (using recursion based on tie-corrected spearman rank correlation) was run to account for the effect of other models. In order to perform group-level inference, the resulting correlation values for each participant were Fisher z-scored (to approximate normality) and tested against a null hypothesis of mean 0 using a one-sample $t$ test. To account for multiple hemispheric comparisons, a Bonferroni-corrected α level of 0.025 was used to determine statistical significance in our main affordance model.

**Representational similarity analysis: Searchlight.** Searchlight analysis was run following the procedure outlined in [119]. The logic of this analysis is that representational similarity analysis can be run within spheres to provide a whole-brain map of multivariate responses to different models.

As before, all analyses were carried out at subject level. Spheres were generated using rsatoolbox for Python v0.2.0 [120], with a radius of 3 voxels (7.5 mm) and a minimum number of surrounding voxels necessary set as 30%. This analysis was run at a whole-brain level.

We followed the exact same approach as for the ROI-based RSA analysis; that is, neural dissimilarity matrices were calculated using crossnobis distance (with residuals from the first-level GLM for noise covariance calculation), and then compared to model matrices using a tie-corrected spearman correlation prior to z-scoring. We excluded the effect of other models using partial correlation.

In order to run group-level analyses, the resulting maps were then converted back into MNI space, and TFCE-based cluster correction [121] was run within the whole-brain. No significant clusters were found at a whole-brain level.

**Representational similarity analysis: Models.** Our main model of interest concerned the similarity of states in terms of whether they allowed the same or different actions. To test this, we used a count-based dissimilarity matrix which indicated the number of shared actions between states (subtracted from the maximum, 2). For example, as states 1 and 4 both allow the same two actions (−2, +2) their dissimilarity is 0; states 1 and 2 only have one shared action (+2), so their predicted dissimilarity is 1. The count-based approach was also used for the two other control models, link distance (the minimum number of steps between two states), and action magnitude (the number of shared actions, independently of the numerical sign).

To control for reaction time, the mean reaction time in probe trials across the entire fMRI session was taken within each subject and condition. The absolute difference in mean reaction time between states was used as a dissimilarity measure in order to construct a participant-specific representational dissimilarity matrix. This was then excluded via the partial correlation approach outlined above. Visual controls, meanwhile, were based upon the distance in 2D Euclidean space between conditions for each subject. To calculate this, the median position in x and y was taken within each trial using a 1 s window centered around the cluster identified on day 1. The median was then taken across trials within each condition, resulting in a single gaze position for each condition. Conditions were then compared using Euclidean distance, following the prediction that closer positions in visual space would equate to more similar neural patterns in the case of retinotopic mapping.

**Analysis of eyetracking data.** The eyetracking data were first cleaned by removing blinks and surrounding data (using automated blink detection carried out by the Eyelink toolbox, and a buffer window of 100 ms) from the time-series. As gaze data was recorded binocularly, the mean position in x and y across eyes was taken for each time point to obtain a single gaze position for each millisecond. Data was then segmented into trials using a window of −500–2500 ms around stimulus onset. Trials of each condition were then grouped together and the median gaze position in x and y was taken for each time point. To account for temporal noise, the data were then minimally smoothed with a Gaussian kernel with a standard deviation of 4.25 ms (roughly corresponding to a full-width half maximum kernel width of 10 ms) and a maximum window of 20 ms within each condition and subject. This resulted in a single time-series in both x and y for each condition, for each subject.

To test for an action-based change in gaze behavior, we compared the directional shift as a function of different states. State 2 allows only positive operations, while state 3 allows only negative operations. We obtained the relative shift by subtracting fixations for state 3 from state 2 in the x axis, for every time point. If the relative position difference is positive, this means an eye movement further right for states which afford positive actions relative to states which afford negative actions.

In order to carry out group-level analyses, we ran a temporal cluster correction using the MNE package [122,123,124] based on a 1-sample $t$ test with 10,000 permutations. As the hypothesis predicted a positive movement, a one-tailed test was used with an alpha level of 0.05.

Finally, we compared the results to performance. To do so, for each participant we identified the median change in gaze position across every time point within the previously-identified significant cluster. This gave a single value which reflected how strongly the directional effect was for a given participant within the cluster. A more positive value indicated a stronger effect. This was then correlated with performance in probe trials (Pearson r) in the first day to obtain a measure of how much gaze effects reflected individual performance. An α level of 0.05 was used.

**DeepMReye.** To gain an estimate of eye position on the second day, the DeepMReye toolbox [65] was used. This toolbox uses a pre-trained CNN to estimate gaze position based on the magnetic resonance signal of the eyeballs.

Following the minimal fMRI preprocessing steps outlined above, gaze positions in x and y (in visual angle) were estimated at a resolution of 10 Hz (10 estimates per TR) using pre-trained model weights based on a task in which participants sequentially fixated different images (Dataset 6).

To gain a single estimate of gaze position within each trial, the median × position per trial was taken within a slightly-extended window of 1 s centered around the cluster identified on the first day (to ensure an even number of datapoints per trial, and in line with the standard DeepMReye pipeline of taking the median within a single TR-length window). Within each subject, the median was then calculated across trials of each condition. To compare gaze lateralization across subjects, a paired $t$ test was run predicting that the median gaze position would be further right for condition 2 than condition 3 (α level of 0.05).

To compare the resulting effects to the MRI data, we firstly compared the extent of gaze lateralization with the predicted neural differences between conditions 2 and 3. To account for differences in noise levels, each RDM was normalized and the distance between conditions 2 and 3 was extracted. This was then correlated with the difference in gaze position for each subject for these conditions (Pearson correlated, α = 0.05). Following this, we then checked in an exploratory analysis for a relation between the magnitude of gaze lateralization and the correlation in the entorhinal cortex with the affordance model (Pearson correlation, α = 0.05).

## Supporting information

**S1 Appendix. Supplementary Materials. Fig A. Test of noise in cross-validated RSA.** Cross-validated neural distances within RDMs should be positive if the resulting matrix is driven by consistent pattern responses across runs. The mean distances were greater than zero in both entorhinal cortex ROIs, suggesting that ROI results were driven by signal

rather than noise. **Fig B. Reliability-based voxel selection.** To establish if our entorhinal effects were robust to voxel-level noise, we carried out a permutation-based version of voxel reliability selection to select voxels which responded consistently to each condition across runs. Reduced ROIs were defined using these voxels, and the affordance effects persisted within them. **Fig C. Comparison of affordance effect in right entorhinal cortex to noise ceiling.** The affordance model correlations were compared to a noise ceiling estimated using across-participant variance. The effect in the right entorhinal cortex was not significantly below the estimated noise ceiling, suggesting another model could not have better explained our neural RDMs. **Fig D. Affordance effect in right entorhinal cortex is not driven by task difficulty.** Affordance effects remained significant after accounting for task difficulty by regressing out a model based on reaction time in probe trials. **Fig E. Distance in visual space correlates with visual but not entorhinal cortex.** (a,b) Neural pattern similarities in visual cortex were correlated with a model based on estimated gaze distance. This was not the case in entorhinal cortex. (c) Moreover, the affordance effect remained significant after controlling for gaze distance using partial correlation. **Fig F. Eye movement differences in the y-axis.** Participants showed a lateralized gaze effect in the y-axis as a function of affordance direction. Unlike in the x-axis, this effect was not correlated with performance. **Fig G. Affordance representation in the motor cortex.** The affordance model was also represented in the right motor cortex, possibly suggesting a task embodiment via motor systems.
(PDF)

## Author contributions

**Conceptualization:** Alexander Eperon, Christian F. Doeller, Stephanie Theves, Roberto Bottini.

**Data curation:** Alexander Eperon.

**Formal analysis:** Alexander Eperon.

**Funding acquisition:** Roberto Bottini.

**Investigation:** Alexander Eperon, Stephanie Theves, Roberto Bottini.

**Methodology:** Alexander Eperon, Stephanie Theves, Roberto Bottini.

**Project administration:** Alexander Eperon, Christian F. Doeller.

**Resources:** Christian F. Doeller, Roberto Bottini.

**Supervision:** Christian F. Doeller, Stephanie Theves.

**Writing – original draft:** Alexander Eperon.

**Writing – review & editing:** Alexander Eperon, Stephanie Theves, Roberto Bottini.

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
