## [Editor Report · Decision Letter 0]

4 Sep 2025

Dear Dr Eperon,

Thank you for submitting your manuscript entitled "Representation of conceptual affordances in eye movements and the entorhinal cortex" for consideration as a Research Article by PLOS Biology and please accept our apologies for the delay in sending you a decision, which was incurred while we sought advice from an Academic Editor with relevant expertise.

Your manuscript has now been evaluated by the PLOS Biology editorial staff as well as by an academic editor with relevant expertise and I am writing to let you know that we would like to send your submission out for external peer review.

Once your full submission is complete, your paper will undergo a series of checks in preparation for peer review. After your manuscript has passed the checks it will be sent out for review. To provide the metadata for your submission, please Login to Editorial Manager (https://www.editorialmanager.com/pbiology) within two working days, i.e. by Sep 08 2025 11:59PM.

Kind regards,

Luke

Lucas Smith, Ph.D.

Senior Editor

PLOS Biology

lsmith@plos.org

---

## [Decision Letter · Decision Letter 1]

30 Oct 2025

Dear Dr Eperon,

Thank you for your patience while your manuscript "Representation of conceptual affordances in eye movements and the entorhinal cortex " was peer-reviewed at PLOS Biology, and I am sorry for the protracted review process. In this case, for some reason it took us a bit longer than normal to get a full set of reviewers signed on, and then I was delayed in moving the decision last week, because I was travelling at a conference. Your study has now been evaluated by the PLOS Biology editors, an Academic Editor with relevant expertise, and by several independent reviewers, and I am writing to let you know that we would like to invite you to revise the work to thoroughly address the reviewers' reports.

The reviews are appended at the end of this email. Note that our Academic Editor has actually provided a full assessment of your paper, and his/her comments are included below as comments from Reviewer 2. The reviewers highlight that this is generally a well conducted and interesting study, but they have suggested a number of ways that the study should be strengthened further, including by refining the presentation and framing of your work.

Given the extent of revision needed, we cannot make a decision about publication until we have seen the revised manuscript and your response to the reviewers' comments. Your revised manuscript is likely to be sent for further evaluation by all or a subset of the reviewers.

**IMPORTANT - SUBMITTING YOUR REVISION**

*Re-submission Checklist*

*Published Peer Review*

*PLOS Data Policy*

*Blot and Gel Data Policy*

Sincerely,

Luke

Lucas Smith, Ph.D.

Senior Editor

PLOS Biology

lsmith@plos.org

REVIEWS:

Reviewer #1: Eperon et al. describe a well-controlled study of learning state transitions. Their focus is characterizing potential signatures of these state transitions in eye movement behaviour and neural representations in entorhinal cortex. Participants first learned hidden structure of transitions between numbers with some transitions forward or backward along the number line more likely than others in a behavioural and eye tracking session. On a second day, participants' recall of the state transitions was tested in a fMRI session. Results show that participants learned, that eye movements during training reflect shifts along a mental number line consistent with state transitions, and that neural representations in entorhinal cortex demonstrate pattern similarity consistent with state similarity. The authors argue that these findings suggest that entorhinal cortex represents actions in a conceptual space which are linked to eye movements.

This study offers compelling results with converging evidence from measures of behaviour, eye movements, and neural similarity. The methods employed are appropriate and well controlled. The statistical checks and supporting analyses are particularly noteworthy as they speak to the reliability of the reported effects. Overall, I find this work valuable and potentially an interesting contribution to the literature. However, several issues should be considered:

1. To be honest, I found the abstract and introduction very confusing. There are many terms and jargon that are bandied about without properly defining them. And, there are many terms offered as equivalents that do not seem to match well. For example, "state transitions" are labelled "affordances", which itself is a very loaded term. Numerical operations are labelled as "actions". The authors refer to state transitions in the experiment's graph structure of numbers as "conceptual affordances". Of course, operational definitions that link constructs to measurable entities is the key to empirical design and the strength of any inference from findings is limited by the scope and appropriateness of operational definitions. In this case, there seems to be a disconnect between the very specific nature of the experiment (i.e., learning non-sequential transitions between numbers) and the targeted constructs. What is a conceptual affordance? And how well do state transitions of +/- 1 or 2 steps along a number line represent such an affordance? The title also feels like an overreach.

2. Two aspects of the existing literature should be given more consideration in the manuscript. First, seminal work showing representations of conceptual spaces according to visual categories in hippocampus is omitted (e.g., studies by Zeithamova, Davis, and Mack) from the introduction and discussion. Second, the successor representation model is only noted in passing (e.g., Stachenfeld et al., 2017). This model is highly relevant to the current study and offers a compelling formal description of how state transitions based on reward/value are learned and encoded. It seems appropriate to give this account more consideration when connecting the current findings to the literature.

3. The different states offer distinct transitions which means there's potentially interesting behavioural effects one might expect. As such, a breakdown of behavioural accuracy for each state would be helpful. Is there a difference in learning among the states? A more detailed look at learning across the states seems especially important in the context of understanding the neural similarity results. For example, if participants had better learning for states with higher conceptual affordance similarity, a further consideration of the RSA results may be warranted.

4. One small point in the discussion: the authors propose that positional representations in entorhinal cortex may be updated by velocity input from eye movements. This is an interesting notion, but it does not seem to be well supported by the control analyses that showed there was no relationship between the distance of gaze position and entorhinal cortex representations. Of course, one cannot strongly argue from a null effect and the authors qualify their proposal with limitations. However, velocity would be reflected in moment-by-moment gaze position, so at least with the given data in the current study, this proposal lacks evidence.

5. Minor type: The text summary of learning performance references figure 1a, which should be figure 2a.

Reviewer #2/Comments from the Academic Editor: Eperon and colleagues describe an fMRI and eye tracking study that measures representations of conceptual affordances in a task where human subjects transition between different 'states' using simple mathematical operations. The main prediction was that the entorhinal cortex would form an abstract representation of the possible transitions (affordances) between different states in a non-spatial task. There are two main findings: (1) eye tracking revealed that subjects tended to look left vs. right when possible affordances were backward on the number line vs forward on the number line, respectively, and (2) the (right) entorhinal cortex represented the affordances that were possible at a given number (state), as defined by representational similarity analyses.

The paper addresses an interesting topic and the predictions are well motivated. The paradigm and analyses are well thought out and the sample of subjects is quite large (over 50 subjects of fMRI data and over 40 for eye tracking). The main results are well motivated and are interesting. The paper is also generally well written and well organized. Supplementary results address some important questions (control analyses, etc.). Overall, my impression of the manuscript is quite positive and I believe the authors have put a lot of thought into the experiment and the writing. The two most significant weaknesses that I see are (1) the findings are somewhat incremental given other recent evidence of conceptual knowledge structures in the hippocampal formation, and (2) most of the core fMRI effects are "just barely" significant. That said, I do not view these as major weaknesses. With respect to the novelty, I do think the current experiment is different—in subtle, but interesting ways—from prior studies. With respect to the strength of findings, I think the results are very well motivated by prior studies (thus, there is somewhat of a 'prior') and I also think the control analyses presented as supplementary figures are very helpful (the control for reaction times is particularly important). Thus, I only have a couple of very minor suggestions.

Major comments

1. Given that there were very substantial reaction time differences between the different states, my biggest concern was that any RSA effects could simply reflect reaction time differences. Supplementary Figure 4 is therefore very important. However, it would be helpful to include a bit more information about exactly how this analysis was implemented. All that is said is that they ran "a control analysis in which we regressed out a model based on average reaction time by state." What was the reaction time model? Was it a similarity matrix based on reaction time similarity? What is reaction time similarity—just the raw difference in mean reaction times? Were reaction times log transformed to correct for non-normality? Would that matter? Again, the reaction time differences between conditions are very large and it certainly seems plausible that this could explain the results.

2. I also wonder whether the RSA results observed in entorhinal might also be observed (perhaps more strongly) in motor cortex? If so, that does not necessarily mean that the entorhinal results reflect task demands, but I think it would be relevant to test a motor region—for the same reason that it is useful to include visual cortex as a comparison in Supplementary Figure 5.

Minor comments

3. Ultimately, I was able to understand the experimental design, but it took a bit of time to figure it out. I think it would be helpful to include a figure or text that very simply illustrates that state 1 = numbers that end in 1 or 6 (e.g., 21, 26), and state 1 = +2 or -2, etc. Again, I was able to figure this out, but it could have been easier.

4. Relatedly: there is a line in the text that says "In particular, eye movements could be expected to skew more rightwards for states that allow only positive actions (e.g. state 2) relative to states that allow only negative actions (e.g. state 3)." The use of "e.g." (for example) is confusing because it implies that there are OTHER states that also allow for only positive actions or only negative actions. But (unless I missed something), it is ONLY state 2 and state 3 that are relevant for this analysis. Thus, it should be "i.e.," instead of "e.g." It's a minor point, but it created some confusion for me.

Reviewer #3: Summary

The authors report results from fMRI and eye tracking experiments (57 and 46 human subjects, respectively). They describe that fMRI voxel patterns in the right entorhinal cortex correlate with similarities of possible actions in a task with mathematical operations. Gaze was correlated with positive versus negative mathematical operations. The authors suggest that these findings indicate that action information is integrated into entorhinal representations of conceptual spaces. This is an interesting manuscript with a range of analyses, which would benefit from a clearer presentation with important information included earlier in the main text. Please find my comments listed below.

Major comments

(1) The introduction and discussion talk about the involvement of the hippocampal formation in mapping relational knowledge, but fMRI RSA focuses on the entorhinal cortex. The rationale for just looking at the entorhinal cortex is unclear. Several MTL regions should be tested to establish the specificity of the effects to the entorhinal cortex.

(2) The task should be better explained in the main text, and the different states should be explained early on. For example, in Fig. 1a, which of the numbers correspond to states 1, 2, 3, and 4 (same for Fig. 1c)? Also, in Fig. 1c, what does S1-4 mean? When the recall task is first mentioned in the main text, it is unclear what it refers to, etc. This opaqueness in task description makes it hard to follow the rest of the results.

(3) Why did you apply slice time correction during fMRI preprocessing given that your TR was 1s?

(4) Why didn't you extract subject-specific ROIs using FreeSurfer? The masks from the Jülich atlas may not closely correspond to the subject's individual anatomy.

(5) The unsignificant whole-brain RSA should be reported in the main text.

(6) It was unclear to me whether it is justified to equate actions, possible transitions, and affordances.

(7) It was also unclear to me whether the term "numerical operations" should be generalized to "actions," as the latter encompasses far more than numerical processes. Moreover, since German students are typically familiar with the concept of a number line, the observed effects may be biased by this prior knowledge.

(8) The authors seem to use the terms cognitive maps (introduction) and cognitive graphs (task description) interchangeably, but I do not believe that this is correct (e.g., Peer et al., TiCS, 2021).

Minor comments

(1) Why did you use Mahalanobis distance for estimating the RDMs? To the best of my knowledge, Pearson correlations are standard. Please explain and provide the results from Pearson correlations.

(2) Was there also a difference in eye tracking y-position between states 2 and 3?

(3) Link distance and action magnitude should be explained in the main text.

(4) "'2' to '5' would be similar to '55' to '57'" seems wrong.

(5) "median change in gaze position across every timepoint within all previously-identified significant clusters" - were there multiple significant clusters? Fig. 3b shows only one cluster.

(6) "which would involve a 4-by-4 correlation" - why is it a 4-by-4 correlation? Aren't six values correlated with each other?

---

## [Decision Letter · Decision Letter 2]

16 Mar 2026

Dear Dr Eperon,

Thank you for your patience while your manuscript "Action in a conceptual space: representation of numerical operations in eye movements and the entorhinal cortex" was peer-reviewed at PLOS Biology. It has now been evaluated by the PLOS Biology editors, an Academic Editor with relevant expertise, and by several independent reviewers.

Based on the reviews, we are likely to accept this manuscript for publication, provided you satisfactorily address the remaining points raised by reviewer 3. Please also make sure to address the following data and other policy-related requests.

**IMPORTANT: Please address the following editorial requests.

1) TITLE: We would like to propose a tweak to the title which we think makes the findings a bit clearer. If you agree, we suggest you change the title to:

'Action information is integrated into entorhinal representations of conceptual space and is reflected in eye movements"

2) ETHICS STATEMENT: Please update the ethics statement in your methods section, to indicate whether the study was conducted according to the principles expressed in the Declaration of Helsinki.

3) DATA AVAILABILITY: I see that your data availability statement currently says 'All relevant data for replications will be uploaded to a publicly available database after acceptance.' Please do upload the data underlying your study to a relevant repository. You can keep this set as 'private' until publication, but I need to check that the dataset meets our data policy (which, which requires that all data be made available without restriction: http://journals.plos.org/plosbiology/s/data-availability.)

Once you have uploaded the underlying data, please also ensure that figure legends in your manuscript include information on where the underlying data can be found, and ensure your supplemental data file/s has a legend. Please update your Data Statement in the submission system to accurately describe where your data can be found.

4) CODE: Thank you for providing the code for your study as a Github link. While this is helpful, please note that we cannot accept sole deposition of code in GitHub, as this could be changed after publication. However, you can archive this version of your publicly available GitHub code to Zenodo. Once you do this, it will generate a DOI number, which you will need to provide in the Data Accessibility Statement (you are welcome to also provide the GitHub access information). See the process for doing this here: https://docs.github.com/en/repositories/archiving-a-github-repository/referencing-and-citing-content

We expect to receive your revised manuscript within two weeks.

*Published Peer Review History*

*Press*

Sincerely,

Luke

Lucas Smith, Ph.D.

Senior Editor

lsmith@plos.org

PLOS Biology

Reviewer #1: I am very happy with the thorough responses and revised manuscript. My initial concerns are well addressed, and I think the revised manuscript is fit for publication.

Reviewer #2/Academic Editor: I only had minor comments / questions in the prior round of review and I am satisfied with the authors' responses. I also found the responses to the other reviewers to be thoughtful. I think this is a strong manuscript that will be of broad interest to readers.

Reviewer #3: The authors have addressed all my comments and I'm fine with their responses. But please do report the null findings regarding PHC and PhG in the manuscript (my previous comment 1).

---

## [Editor Report · Decision Letter 3]

27 Mar 2026

Dear Dr Eperon,

Thank you for the submission of your revised Research Article "Action information is integrated into entorhinal representations of conceptual space and is reflected in eye movements" for publication in PLOS Biology, and thank you also for addressing the last reviewer and editorial requests in this revision. On behalf of my colleagues and the Academic Editor, Brice Kuhl, I am pleased to say that we can in principle accept your manuscript for publication, provided you address any remaining formatting and reporting issues. These will be detailed in an email you should receive within 2-3 business days from our colleagues in the journal operations team; no action is required from you until then. Please note that we will not be able to formally accept your manuscript and schedule it for publication until you have completed any requested changes.

PRESS

Sincerely,

Luke

Lucas Smith, Ph.D.

Senior Editor

PLOS Biology

lsmith@plos.org